# Interfacial compatibility critically controls Ru/TiO$_2$ metal-support interaction modes in CO$_2$ hydrogenation

Jun Zhou[1,3], Zhe Gao [2,3], Guolei Xiang [1✉], Tianyu Zhai[1], Zikai Liu[1], Weixin Zhao[1], Xin Liang [1] & Leyu Wang [1✉]

Supports can widely affect or even dominate the catalytic activity, selectivity, and stability of metal nanoparticles through various metal-support interactions (MSIs). However, underlying principles have not been fully understood yet, because MSIs are influenced by the composition, size, and facet of both metals and supports. Using Ru/TiO$_2$ supported on rutile and anatase as model catalysts, we demonstrate that metal-support interfacial compatibility can critically control MSI modes and catalytic performances in CO$_2$ hydrogenation. Annealing Ru/rutile-TiO$_2$ in air can enhance CO$_2$ conversion to methane resulting from enhanced interfacial coupling driven by matched lattices of RuO$_x$ with rutile-TiO$_2$; annealing Ru/anatase-TiO$_2$ in air decreases CO$_2$ conversion and converts the product into CO owing to strong metal-support interaction (SMSI). Although rutile and anatase share the same chemical composition, we show that interfacial compatibility can basically modify metal-support coupling strength, catalyst morphology, surface atomic configuration, MSI mode, and catalytic performances of Ru/TiO$_2$ in heterogeneous catalysis.

[1] State Key Laboratory of Chemical Resource Engineering, Beijing University of Chemical Technology, Beijing 100029, China. [2] State Key Laboratory of Coal Conversion, Institute of Coal Chemistry, Chinese Academy of Sciences, Taoyuan South Road 27, Taiyuan 030001, China. [3] These authors contributed equally: Jun Zhou, Zhe Gao. ✉email: xianggl@mail.buct.edu.cn; lywang@mail.buct.edu.cn

Supported metal nanoparticles (NPs) dominate practical catalysts in producing bulk and fine chemicals, reducing environmental emissions, energy conversions, etc.[1–3]. Their catalytic performances (activity, selectivity, and stability) not only depend on the composition, size, shape, and ligation state of metal NPs, but are also highly affected or even dominated by supports[4–11]. Metal-support interaction (MSI) has therefore become a central topic in heterogeneous catalysis[1,3,12,13]. Typical MSI modes include strong metal-support interaction (SMSI), interfacial charge transfer, interfacial perimeter, spillover, etc.[1]. In particular, SMSI phenomena, featuring in encapsulating metallic NPs (such as Pt, Au, Pd, Rh, Ru, and Ni) by reducible oxide supports like TiO$_2$, can widely modify catalytic activity and selectivity of hydrogenation reactions[9,14–20]. Although various MSI forms have been reported, the underlying tuning principles still remain elusive, because all structural factors can affect MSIs, such as the composition, size, and shape of metals, the composition, phase, facet, and size of supports, as well as adsorbates and reaction atmospheres[1,12,21–25]. Moreover, their simultaneous interactions extremely complicate MSI phenomena and challenge the study on the mechanisms. Exploring the principles dominating support effects and MSIs is therefore crucial for rational design, optimization, and understanding of heterogeneous catalysis.

Among all structural factors, metal-support interface should play the primary role, because all MSI modes occur based on the direct contacts of catalysts with supports[26,27]. For solid–solid contacts, coupling strength is the most fundamental parameter determining the property and stability of their interfaces, which is thermodynamically described with adhesion energy ($\Phi_{adh}$)[28,29]. The relative value of $\Phi_{adh}$ to bulk cohesion energy ($\Phi_{coh}$) basically determines interfacial contact angles and thermal stability of supported particles[30]. In catalysis science, adhesion energy widely controls the morphologies and sintering rates of supported-metal NPs[31,32]. Many post-treatment methods such as thermal

annealing and reduction–oxidation cycles can modify interfacial adhesion and catalytic performances[1,9,16,22,33,34]. Furthermore, at the atomic scale, interfacial coupling occurs through forming chemical bonds, thus, interfacial bonding strength basically determines metal-support adhesions[35]. For example, Campbell et al. theoretically studied the trends in the adhesion energies of metal NPs on various oxide surfaces, and found that higher metal oxophilicity and more active surface-oxygen atoms could lead to stronger metal-support adhesions[29]. Senftle et al. further revealed that interfacial binding strengths between single-metal atoms and oxide supports depended on the oxophilicity of supported metals and reducibility of oxide supports[36]. Despite these understandings on the surface stability of metal NPs on oxides, the structure–function relationships on how interfacial structure features modify MSI modes and catalytic performances are still not fully revealed yet[9,33,34,37].

At the atomic scale, strong catalyst-support contacts result from interfacial bonds. The strength parameters, macroscale adhesion energy, and microscale bonding energy can be correlated following:

$$\Phi_{adh} = kE_{IB}N_s \qquad (1)$$

where $E_{IB}$ and $N_s$ denote the average energy and surface density of interfacial bonds, and $k$ is a coefficient. Thus, $\Phi_{adh}$ can be enhanced by increasing $E_{IB}$ or $N_s$. Both $E_{IB}$ and $N_s$ further depend on the atomic configurations of contacting surfaces, because the positions of interfacial atoms intrinsically affect the length, angle, and density of interfacial bonds. Therefore, interfacial configurations of catalysts and supports are intrinsic structural factors controlling catalyst states and MSI effects. The matching degree of interfacial configurations is also referred to as interfacial compatibility, which measures the strength of interfacial bonding and adhesion[27,30,35]. Catalysts weakly wet supports at misfit interfaces with low interfacial compatibilities, which leads to ready phase separations at the interfaces and catalyst sintering.

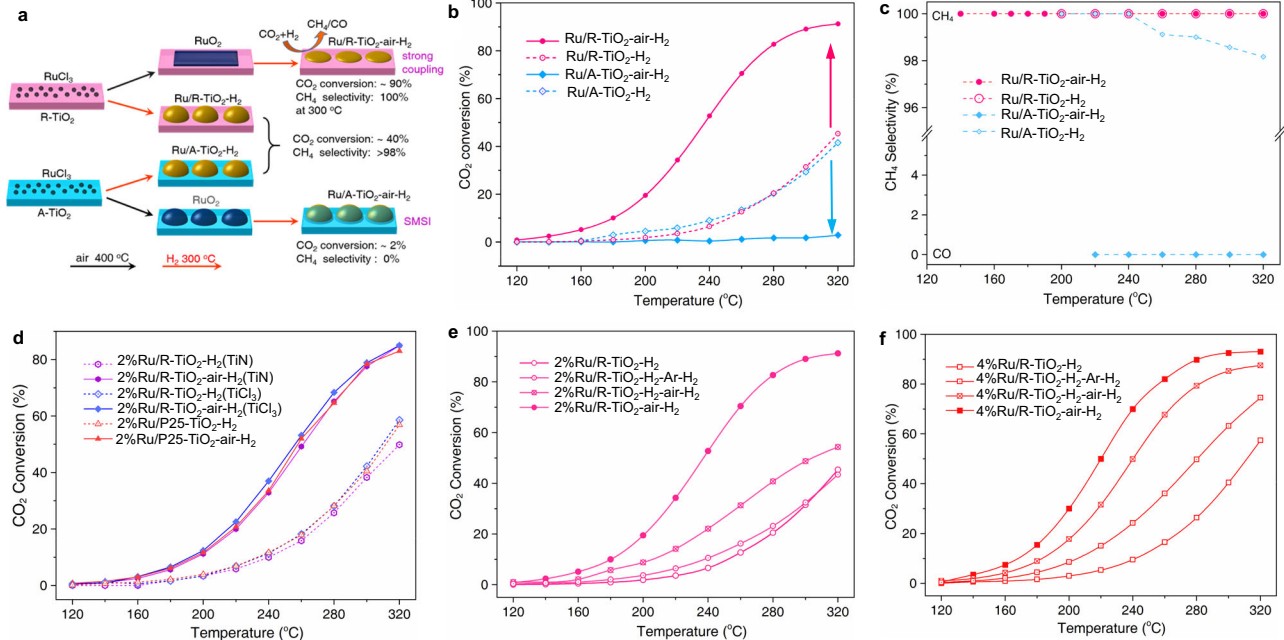

**Fig. 1 Opposite catalytic performances of Ru/TiO$_2$ supported on rutile (R–TiO$_2$) and anatase (A–TiO$_2$) for CO$_2$ hydrogenation. a** Summary scheme of varied activity and selectivity by direct H$_2$ reduction and after annealing in air. Ru/TiO$_2$–H$_2$ refers to directly reduced catalysts by H$_2$, while Ru/TiO$_2$–air–H$_2$ refers to catalysts by annealing in air at 400 °C and further reduction by H$_2$. **b** Temperature-dependent CO$_2$ conversions. **c** Temperature-dependent CH$_4$ selectivity. **d** Temperature-dependent CO$_2$ conversions of 2% Ru/R–TiO$_2$ catalysts on other R–TiO$_2$ supports that were prepared from TiN and TiCl$_3$, and P25 TiO$_2$. Temperature-dependent CO$_2$ conversions by annealing (**e**) 2% Ru/R–TiO$_2$–H$_2$ and (**f**) 4% Ru/R–TiO$_2$–H$_2$ catalysts in Ar and air. Ru/R–TiO$_2$–H$_2$–air–H$_2$ means the catalyst was first reduced by H$_2$ at 300 °C, then annealed in air at 400 °C, and reduced with H$_2$ at 300 °C again.

While high interfacial compatibility can increase both the strength and density of interfacial bonds[13,35]. The highest interfacial compatibility exists in epitaxial interfaces between lattice-matched materials. In general, lattice misfit less than 5% can form epitaxial overlayers with zero contact angles. This principle has widely guided the growth of semiconductor and oxide heterogeneous structures with minimized interfacial defects[38]. However, the mechanism of how metal-support contacts and their interfacial compatibilities affect MSI modes and catalytic performances has been rarely revealed yet.

Using $Ru/TiO_2$ as a model catalyst, here we demonstrate the critical role of interfacial compatibility in controlling MSI modes, surface atomic states, and catalytic activity and selectivity in $CO_2$ hydrogenation reaction. Rutile ($R–TiO_2$) and anatase ($A–TiO_2$) are used as supports to vary interfacial structures. $RuO_2$ shares the same lattice structure with $R–TiO_2$, which can lead to a high interfacial compatibility to form epitaxial overlayers[39]. The interfacial $RuO_x$ species can act as anchoring layers to strengthen interfacial bonding between Ru and $R–TiO_2$. For $CO_2$ hydrogenation reaction, we find that both catalytic activity and selectivity can be oppositely modified by annealing $Ru/TiO_2$ catalysts in air. On $R–TiO_2$, Ru shows enhanced activity and dominant selectivity of $CH_4$; on $A–TiO_2$, Ru shows decreased activity and dominant selectivity of CO (Fig. 1a). Such opposite catalytic performances clearly indicate that interfacial compatibility can vary MSI modes—$Ru/A–TiO_2$ shows normal SMSI effect, while $Ru/R–TiO_2$ displays strong interfacial coupling.

## Results

### Support effects on activity and selectivity of $Ru/TiO_2$ for $CO_2$ hydrogenation.

All $Ru/TiO_2$ catalysts were prepared following the same protocol to minimize the interference from synthetic conditions. To stabilize the surfaces and prevent further sintering during catalyst preparations and reactions, the supports were first annealed in air at 500 °C for 10 h (Supplementary Fig. 1). The annealed supports and $RuCl_3$ were uniformly mixed in water, rapidly frozen with liquid nitrogen, and dried in a freezing drier (Supplementary Fig. 2). The fully mixed $RuCl_3–TiO_2$ precursors were reduced by $H_2$ at 300 °C either directly ($Ru/TiO_2–H_2$) or after pre-annealing in air at 400 °C ($Ru/TiO_2–air–H_2$). $CO_2$ hydrogenation was conducted at normal pressure with gas hourly space velocity (GHSVs) of 12,000 $mL·g^{-1}·h^{-1}$, in which the reaction gas composed of 60 vol% $H_2$/15 vol% $CO_2$/25 vol% Ar. At normal pressure, the products of $CO_2$ hydrogenation are $CH_4$ and CO. Therefore, $CO_2$ conversion denotes catalytic activity, and the ratio of $n(CH_4)/(n(CO) + n(CH_4))$ denotes selectivity, while the varied activity and selectivity further reflect different support effects and MSI modes[5,12,40–43].

Annealing $RuCl_3–TiO_2$ precursors in air can effectively modify catalytic performances of $Ru/TiO_2$ catalysts (Fig. 1a, b). Figure 1b presents temperature-dependent $CO_2$ conversions by 2% $Ru/TiO_2$. $Ru/R–TiO_2–H_2$ and $Ru/A–TiO_2–H_2$ show similar $CO_2$ conversions between 120 °C and 320 °C, suggesting that $R–TiO_2$ and $A–TiO_2$ apply similar support effects on directly reduced $Ru/TiO_2$ catalysts. However, the conversions dramatically differentiate by pre-annealing $RuCl_3–TiO_2$ precursors in air at 400 °C. $Ru/R–TiO_2–air–H_2$ displays an enhanced catalytic performance, with $CO_2$ conversion at 300 °C increasing from 31.4% to 89.2%; $Ru/A–TiO_2–air–H_2$ shows a highly decreased activity, with $CO_2$ conversion at 300 °C reducing from 29.4% to 1.7%. At each reaction temperature, $Ru/R–TiO_2–air–H_2$ shows the highest $CO_2$ conversions among four catalysts. The results indicate that $R–TiO_2$ and $A–TiO_2$ apply opposite support effects on the activity of Ru NPs.

In addition to activity, annealing $RuCl_3–TiO_2$ precursors in air can also dramatically modify the catalytic selectivity of Ru on

$R–TiO_2$ and $A–TiO_2$ (Fig. 1c). $CH_4$ dominates both the products of $Ru/R–TiO_2–H_2$ and $Ru/A–TiO_2–H_2$ between 200 °C and 320 °C (>98%). However, the product on $Ru/A–TiO_2–air–H_2$ converts into 100% CO between 220 °C and 320 °C, while on $Ru/R–TiO_2–air–H_2$ is still 100% $CH_4$ between 140 °C and 320 °C. The different products show another effect of supports on the catalytic performances of Ru NPs[18,41].

The enhancement effect of air-annealing on the activity of $Ru/R–TiO_2$ is a general trend. We verified the phenomena by varying the loading amounts of Ru, rutile supports, and post-processing procedures. The activities of 1% $Ru/R–TiO_2$ and 4% $Ru/R–TiO_2$ can also be enhanced by pre-annealing in air, showing the same trend with 2%-$Ru/R–TiO_2$ (Supplementary Fig. 3). Moreover, the activities of 2% $Ru/R–TiO_2$ supported on other $R–TiO_2$ materials prepared using $TiCl_3$ and TiN (Fig. 1d) as precursors, and P25 $TiO_2$, a commercial $TiO_2$ product composed of 4/5 anatase and 1/5 rutile, can also be effectively enhanced by pre-annealing in air. It is noted that $CO_2$ conversions by $Ru/R–TiO_2–H_2$ and $Ru/R–TiO_2–air–H_2$ are almost the same on these three supports, confirming the stable reproducibility of this enhancement effect.

We further annealed 2% $Ru/R–TiO_2–H_2$ (Figs. 1e) and 4% $Ru/R–TiO_2–H_2$ (Fig. 1f) catalysts at 400 °C in 25-sccm air or Ar flows, and then reduced with $H_2$ at 300 °C. $CO_2$ conversion by the annealed catalyst in Ar (2% $Ru/R–TiO_2–H_2–Ar–H_2$) is similar to that of 2% $Ru/R–TiO_2–H_2$. In contrast, $CO_2$ conversions by the annealed catalyst in air (2% $Ru/R–TiO_2–H_2–air–H_2$) notably increase between 160 °C and 320 °C. In particular, $CO_2$ conversion increases from 1.9% to 8.8% at 200 °C, and 31.5% to 48.7% at 300 °C. This promotion effect can also be supported by the reduced apparent activation energies ($E_a$, Supplementary Fig. 4). $E_a$ of 2% $Ru/R–TiO_2–H_2$ is 66.5 $kJ·mol^{-1}$, while the values are 52.3 $kJ·mol^{-1}$ and 40.4 $kJ·mol^{-1}$ for 2% $Ru/R–TiO_2–H_2–Ar–H_2$ and 2% $Ru/R–TiO_2–H_2–air–H_2$, respectively[5]. The enhancement effect on 4% $Ru/R–TiO_2–H_2$ is more apparent than 2% $Ru/R–TiO_2$ between 140 °C and 320 °C after annealing in Ar and air. This is because Ru nanoparticles on 4% $Ru/R–TiO_2–H_2$ are larger, and annealing can more effectively increase their contacts with $R–TiO_2$ supports.

### Geometric states of Ru NPs on $TiO_2$ supports.

Given that $R–TiO_2$ and $A–TiO_2$ share the same chemical compositions, supports and catalysts were annealed and prepared following the same procedures, and reaction conditions were controlled the same, $Ru–TiO_2$ interfacial interactions should dominate the opposite support effects. $RuO_2$ shares the same lattice structure with $R–TiO_2$, and their lattice misfit is less than 3.0%, thus $RuO_2$ can form epitaxial overlayers on $R–TiO_2$ with zero contact angles[39]. Figure 2a, b and Supplementary Fig. 5 present transmission electron microscopy (TEM) and scanning TEM (STEM) images of $RuO_2/R–TiO_2$ prepared by annealing $RuCl_3–R–TiO_2$ mixture at 400 °C in air. $RuO_2$ encapsulates $R–TiO_2$ nanorods as epitaxial overlayers and forms core–shell structures. Such epitaxial structures can enhance the activity and stability of $RuO_2$ in catalytic oxidation reactions, such as Deacon reaction[44]. In contrast, $RuO_2$ supported on $A–TiO_2$ are NPs rather than epitaxial overlayers due to their different lattice types (Supplementary Fig. 6). Therefore, $RuO_2$ shows a much higher interfacial compatibility with $R–TiO_2$ than $A–TiO_2$—this intrinsically determines their opposite support effects on the catalytic performances of Ru NPs.

The different interfacial compatibilities modify the surface states of Ru NPs on $TiO_2$ supports first. For $Ru/R–TiO_2–air–H_2$, Ru can still partly show epitaxial structures after $H_2$ reduction (Fig. 2c). In particular, after $CO_2$ hydrogenation reaction, Ru presents as flat NPs on $Ru/R–TiO_2–air–H_2$ as shown by the

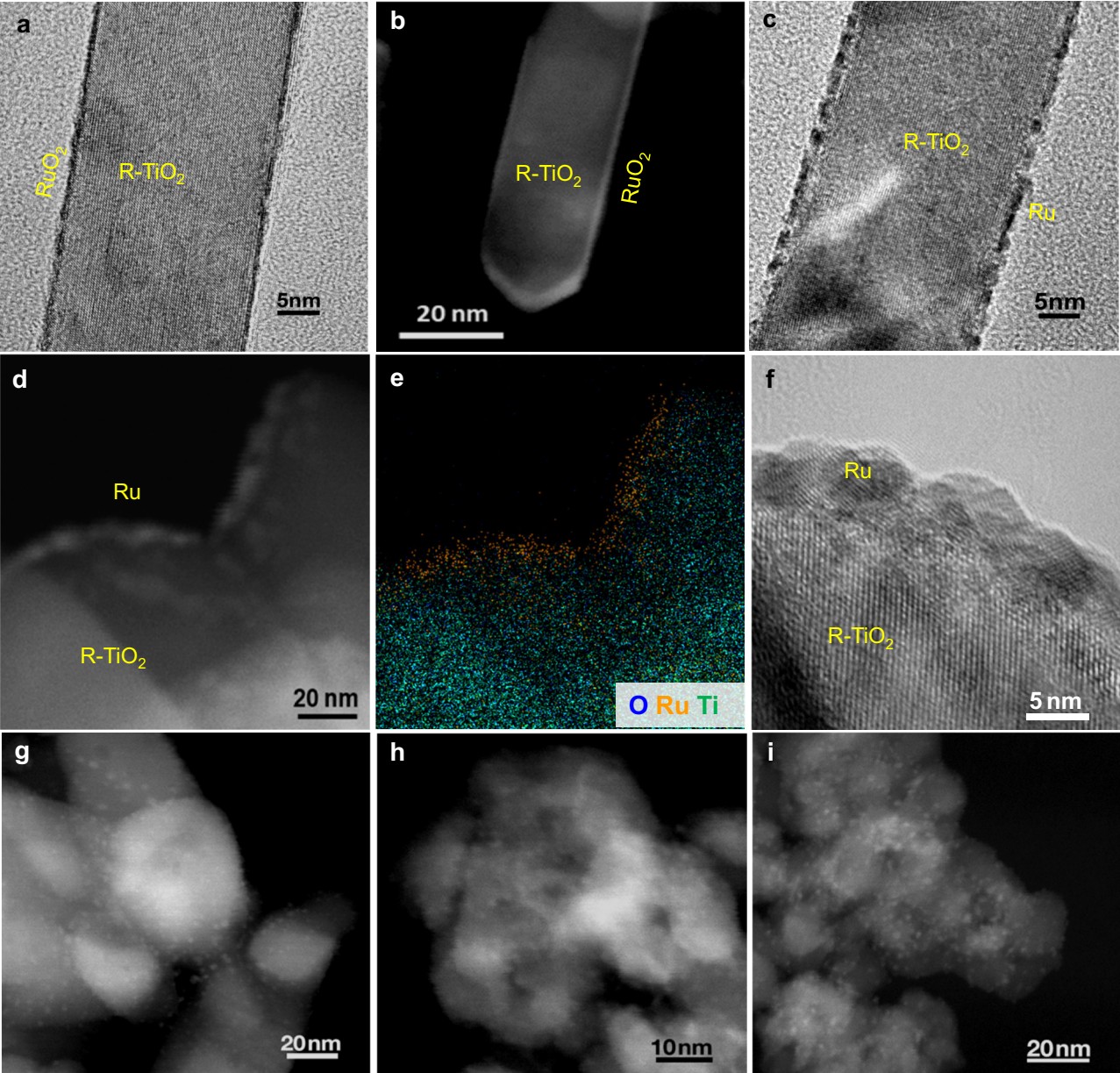

**Fig. 2 Structure characterizations of Ru/TiO$_2$ catalysts. a** TEM and **b** STEM images of RuO$_2$ overlayers on R–TiO$_2$ nanorods. **c** Ru/R–TiO$_2$ reduced from RuO$_2$/R–TiO$_2$ by H$_2$. **d** STEM, **e** elemental mapping, and **f** TEM images of Ru/R–TiO$_2$–air–H$_2$ after CO$_2$ hydrogenation. STEM images of **g** Ru/R–TiO$_2$–H$_2$, **h** Ru/A–TiO$_2$–air–H$_2$, and **i** Ru/A–TiO$_2$–H$_2$ after reactions.

images of STEM (Fig. 2d), elemental mapping (Fig. 2e), and high-resolution TEM (Fig. 2f). In contrast, for Ru/ R–TiO$_2$–H$_2$, Ru/A–TiO$_2$–H$_2$, and Ru/A–TiO$_2$–air–H$_2$, Ru exists as NPs with average sizes around 2.0–3.0 nm after reactions (Fig. 2g, i, Supplementary Fig. 7). Specifically, size distributions of Ru nanoparticles on Ru–TiO$_2$–air–H$_2$ and Ru–TiO$_2$–H$_2$ are 2.4 ± 0.4 nm and 2.5 ± 0.5 nm, respectively (Supplementary Fig. 7). The high interfacial compatibility can intrinsically increase interfacial coupling strength, and modify the chemical and surface states of Ru species, MSI modes, and catalytic performances of Ru/TiO$_2$ catalysts.

**Interfacial bonding states of Ru/TiO$_2$ catalysts**. We use H$_2$ temperature-programmed reduction (H$_2$-TPR) to probe the effects of interfacial compatibility on Ru–TiO$_2$ coupling strengths (Fig. 3a). H$_2$-TPR is an effective method to characterize the reducibility of oxides and their interfacial interaction strengths

with supports[37,45]. We prepared RuO$_2$/TiO$_2$ materials by annealing RuCl$_3$–TiO$_2$ mixtures at 400 °C in air. H$_2$-TPR results show that RuO$_2$/A–TiO$_2$ can be reduced between 110 °C and 175 °C; the peaks at 128 °C, 150 °C, and 300 °C correspond to surface RuO$_2$, interfacial RuO$_x$, and surface A–TiO$_2$ species, respectively. While RuO$_2$/R–TiO$_2$ can be reduced between 100 °C and 290 °C, and shows three states at 138, 185, and 270 °C, corresponding to surface RuO$_2$, interfacial RuO$_x$ species[45]. The higher reduction temperature indicates that RuO$_x$ is more stable on R–TiO$_2$, which further confirms the stronger interfacial coupling due to matched lattices. Furthermore, X-ray diffraction (XRD) patterns of Ru/A–TiO$_2$ display clear Ru signals, while no Ru peaks appear on Ru/R–TiO$_2$–air–H$_2$ (Fig. 3b, Supplementary Fig. 8). The results indicate higher dispersions of Ru on R–TiO$_2$, agreeing with TEM results.

We use X-ray photoelectron spectroscopy (XPS) to probe the chemical states of Ru. Figure 3c presents Ru-*3d* and C-*1s* lines

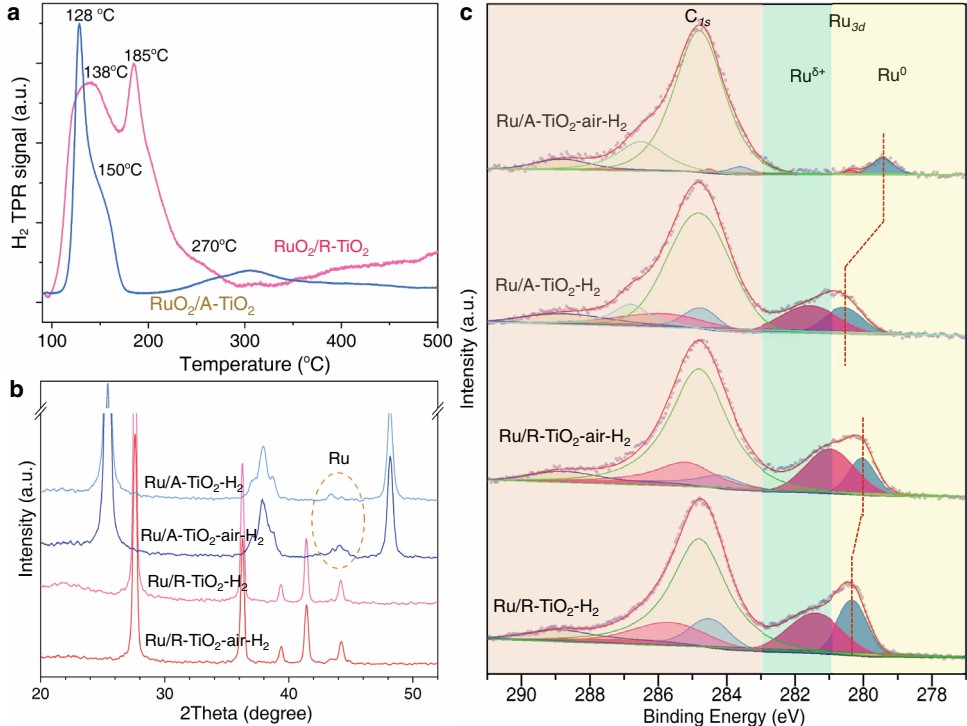

**Fig. 3 Characterizing the chemical states of Ru species on TiO$_2$ supports. a** H$_2$-TPR of RuO$_2$/A–TiO$_2$ and RuO$_2$/R–TiO$_2$. **b** XRD patterns and **c** XPS spectra of Ru/TiO$_2$ catalysts.

**Table 1 XPS results of 2% Ru/TiO$_2$ catalysts.**

| Sample | Binding energy (eV) | | Ru$^{\delta+}$/Ru$^0$ radio |
|---|---|---|---|
| | Ru$^{\delta+}$ | Ru$^0$ | |
| Ru/R–TiO$_2$–air–H$_2$ | 281.0 | 280.0 | 2.2 |
| Ru/R–TiO$_2$–H$_2$ | 281.4 | 280.3 | 1.3 |
| Ru/A–TiO$_2$–air–H$_2$ | 280.3 | 279.4 | 0.2 |
| Ru/A–TiO$_2$–H$_2$ | 281.5 | 280.5 | 1.5 |

of 2% Ru/TiO$_2$ catalysts after catalytic reactions, in which C-1s is set at 284.8 eV. Ru-3$d_{3/2}$ spectra show both metallic (Ru$^0$) and oxidized (Ru$^{\delta+}$) states, in which oxidized states mainly locate at Ru–TiO$_2$ interfaces. Ru$^{\delta+}$/Ru$^0$ ratio is 1.3 for Ru/R–TiO$_2$–H$_2$ and 1.5 for Ru/A–TiO$_2$–H$_2$ (Table 1). The approximately identical Ru$^{\delta+}$/Ru$^0$ ratios suggest that directly-reduced Ru NPs show similar interactions with R–TiO$_2$ and A–TiO$_2$, agreeing with their catalytic performances. However, pre-annealing in air can dramatically alter the states of Ru. Ru$^{\delta+}$/Ru$^0$ ratio increases to 2.2 for Ru/R–TiO$_2$–air–H$_2$, indicating increased interfacial contacts. This trend can be reproduced on other Ru/R–TiO$_2$ catalysts of varied R–TiO$_2$ supports and loading ratios of Ru (Supplementary Figs. 9 and 10, Supplementary Table 1). In contrast, Ru$^{\delta+}$/Ru$^0$ ratio decreases to 0.2 for Ru/A–TiO$_2$–air–H$_2$, and Ru$^{\delta+}$ state is very weak. The opposite trends of Ru$^{\delta+}$/Ru$^0$ ratio also agree with their opposite catalytic performances.

Another feature of Ru/A–TiO$_2$–air–H$_2$ lies in the shift of Ru$^0$-3$d_{5/2}$ from 280.5 to 279.4 eV, a binding energy even lower than that of Ru foil (280.1 eV, Supplementary Fig. 11). This phenomenon is usually ascribed to the occurrence of SMSI effect[46]. This can also be supported by the catalytic performances. For CO$_2$ hydrogenation, SMSI and catalyst sizes can

highly affect activity and selectivity[5,21,22,45]. Both SMSI effect and size reduction can convert the product from CH$_4$ to CO[43,47,48]. For example, Paraskevi reported that 3-nm Ru NPs showed the highest turnover frequency (TOF) on TiO$_2$, and bigger NPs favor CH$_4$[5]. In our system, pre-annealing increases the size of Ru NPs on A–TiO$_2$ from 2.1 nm to 2.8 nm, but the activity and CH$_4$ selectivity both dramatically decrease. The results indicate that SMSI effect should account for this change, because SMSI can generally decrease the activity and selectivity of CO$_2$ methanation[18]. For example, for Rh/TiO$_2$, adsorbates can induce SMSI to decrease the activity and selectivity for CO$_2$ methanation; Ru/TiO$_2$ also shows SMSI effect in reduction reactions[18,49,50]. Therefore, varied interfacial compatibilities can vary MSI modes of Ru/TiO$_2$–air–H$_2$ catalysts: Ru/R–TiO$_2$–air–H$_2$ shows enhanced interfacial coupling bridged by RuO$_x$ layers, while Ru/A–TiO$_2$–air–H$_2$ shows SMSI effect.

**Surface atomic states of Ru NPs probed with CO-DRIFTS.** Different MSI modes can vary the exposed surface atomic states of Ru NPs. We characterize surface Ru sites using diffuse-reflectance infrared Fourier transform spectroscopy (DRIFTS) at 25 °C with CO as the probe (Fig. 4). Figure 4a schemes the possible adsorption configurations of CO on Ru/TiO$_2$. For Ru/A–TiO$_2$–H$_2$, 2140 and 2080 cm$^{-1}$ result from multi-carbonyl-adsorption modes of CO on Ru sites with low coordination numbers (Ru(CO)$_x$, $x = 2, 3$), while the modes appear at 2138 cm$^{-1}$ and 2075 cm$^{-1}$ for Ru/R–TiO$_2$–H$_2$[51]. The broad peaks from 1900 to 2070 cm$^{-1}$ result from top-absorption modes of CO on Ru NPs (Ru–CO) and at the interface between Ru and TiO$_2$ (Ru$_{if}$–CO)[41,42]. For Ru/R–TiO$_2$–air–H$_2$, the proportion of 1950 derived from Ru$_{if}$–CO was higher than directly reduced, agreeing with the XPS results. The bands at 2030 and 2015 cm$^{-1}$ belong to CO linearly adsorbed on Ru surfaces with A–TiO$_2$ and R–TiO$_2$ (Ru–CO), while the peak at 2105 cm$^{-1}$ on A–TiO$_2$ corresponds to Ru nanoclusters (Ru$^{n+}$–CO).

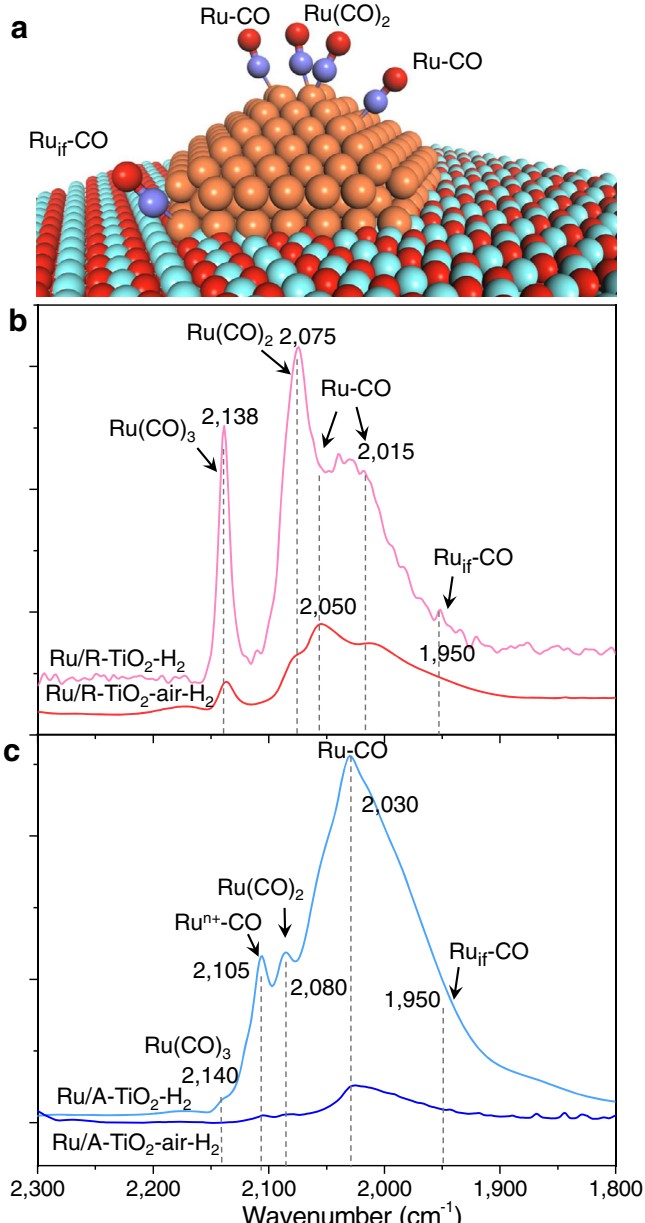

**Fig. 4 Probing surface atomic states of Ru NPs with CO-DRIFTS at 25 °C.**
**a** Adsorption configurations of CO on Ru/TiO$_2$ catalysts. **b** and **c** CO-DRIFTS of Ru/TiO$_2$ catalysts.

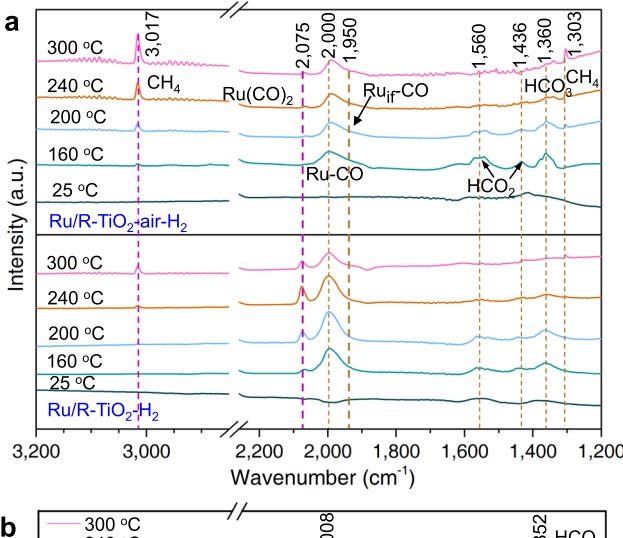

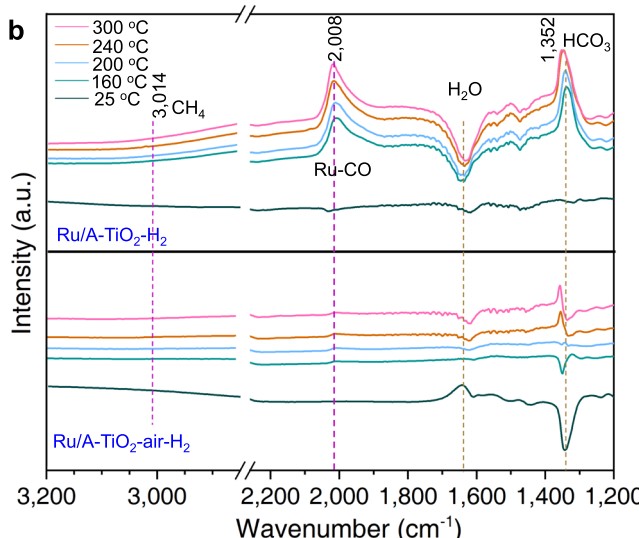

**Fig. 5 Revealing reaction mechanisms of CO$_2$ hydrogenation on Ru/TiO$_2$ catalysts with operando FTIR from 25 °C to 300 °C. a** Reaction mechanisms on Ru/R-TiO$_2$-H$_2$ and Ru/R-TiO$_2$-air-H$_2$. **b** Reaction mechanisms on Ru/A-TiO$_2$-H$_2$ and Ru/A-TiO$_2$-air-H$_2$.

These Ru nanoclusters catalyze the formation of CO (<2%) above 240 °C (Fig. 1c).

After pre-annealing, the total intensities decrease, and the relative intensities of the peaks also change. In particular, Ru(CO)$_x$ mode on low coordinated Ru (Ru$_{LC}$) sites decreases for Ru/R–TiO$_2$–air–H$_2$, while it almost disappears for Ru/A–TiO$_2$–air–H$_2$. These changes mean that the ratios of Ru$_{LC}$ sites decrease after annealing. Usually, the ratio of low coordinated surface atoms decreases with increased particle sizes. The result of Ru/A–TiO$_2$–air–H$_2$ is consistent with this trend, as shown by increased sizes of Ru NPs from 2.1 nm to 2.8 nm (Supplementary Fig. 7). The projected sizes of Ru NPs for Ru/R–TiO$_2$–air–H$_2$ and Ru/R–TiO$_2$–H$_2$ are 2.4 ± 0.4 nm and 2.5 ± 0.5 nm (Supplementary Fig. 7), which are almost the same. This suggests that the decreased ratios of Ru$_{LC}$ sites did not result from increased particle sizes. TEM images show that Ru NPs display flatter shapes for Ru/R–TiO$_2$–air–H$_2$, owing to the

stronger affinity of Ru with rutile supports bridged by interfacial RuO$_x$ species (Fig. 2d–f). The greater curvature radius of flatter particles can lead to more ordered arrangement of surface atoms and increase the number of surrounding atoms. We also characterize the metal dispersion of Ru NPs with CO pulse adsorption. Metal dispersions of Ru ($D_{co}$) are 33.6% and 31.1% for Ru/R–TiO$_2$–air–H$_2$ and Ru/R–TiO$_2$–H$_2$, respectively, which are almost the same (Supplementary Table 2). The results thus indicate that size and surface area does not play critical roles in enhancing the catalytic performances of Ru NPs on R–TiO$_2$ supports. Therefore, such different surface atomic configurations of Ru nanoparticles directly modify their catalytic performances in CO$_2$ methanation.

**Reaction mechanisms probed with operando FTIR.** We further use operando Fourier transform-infrared spectroscopy (operando FTIR) to reveal CO$_2$ hydrogenation mechanisms on these Ru/TiO$_2$ catalysts (Fig. 5). The measurements were performed from 25 to 300 °C in 20-sccm gas flows of 60 vol%H$_2$/15 vol%CO$_2$/25 vol% Ar. For Ru/R–TiO$_2$ (Fig. 5a), the absorptions at 3017 and 1303 cm$^{-1}$ result from C to H bonds of CH$_4$, 1436 and 1560 cm$^{-1}$ from adsorbed formate species (*HCO$_2$), 1950, 2075 from

adsorbed CO species (*CO), and 1360 cm$^{-1}$ from adsorbed carbonate (*HCO$_3$)[5,41,42,52].

At 25 °C, both Ru/R–TiO$_2$–air–H$_2$ and Ru/R–TiO$_2$–H$_2$ show weak adsorption peaks, but the reactions can obviously occur above 160 °C. At 160 °C, *HCO$_3$, *HCO$_2$, and *CO appear on both catalysts, meaning CO$_2$ is first activated as *HCO$_3$ and *HCO$_2$, and further reduced into *CO[8,41]. The difference lies in 2075 and 3017 cm$^{-1}$. At 160 °C, CH$_4$ appears on Ru/R–TiO$_2$–air–H$_2$ but not on Ru/R–TiO$_2$–H$_2$, indicating that Ru/R–TiO$_2$–air–H$_2$ is more active for CO$_2$ methanation. This agrees with the enhanced catalytic results. Many researches have figured out that *CO and formate are two possible intermediates in thermal CO$_2$ hydrogenation reactions. In our results, stepwise-increasing reaction temperatures can lead to similar changes of *HCO$_2$ on the two samples (Fig. 5a). This suggests that formate is not likely the intermediate, or at least not linked to the distinctly different activity on rutile. Some previous reports have also concluded that *CO hydrogenation is the rate-determining step in the CO$_2$ hydrogenation on Ru/TiO$_2$[41]. Two catalysts show different adsorption modes of *CO, more obvious multi-carbonyl Ru(CO)$_n$ species at 2070 cm$^{-1}$ on Ru/R–TiO$_2$–H$_2$, which adsorbed on Ru$_{LC}$ of the surfaces of Ru nanoparticles. It is inactive at low temperatures because H$_2$ cannot effectively reduce it at low temperatures[51].

For Ru/A–TiO$_2$ catalysts, the modes at 3014 cm$^{-1}$ result from C to H bonds of CH$_4$, 2008 cm$^{-1}$ from *CO, and 1352 cm$^{-1}$ from *HCO$_3$ (Fig. 5b). This suggests that the reaction routes are the same with Ru/R–TiO$_2$, agreeing with our catalytic results in fixed-bed reactors. While the reversal peaks at 1644 and 1344 cm$^{-1}$ originate from the desorption of *H$_2$O, *OH, and *H on catalyst surfaces during the reaction[53]. For Ru/A–TiO$_2$–air–H$_2$, the intensity at 2008 cm$^{-1}$ sharply drops compared with Ru/A–TiO$_2$–air–H$_2$, in line with CO-DRIFTS results in Fig. 4c. The highly decreased Ru–CO mode at 2008 cm$^{-1}$ indicates reduced exposure of surface Ru sites, which further supports the occurrence of SMSI, and agrees with the low catalytic activity (Fig. 1b). While SMSI effect can effectively convert the product from CH$_4$ into CO[12,17,18,48].

## Discussion

Our results show that rutile and anatase TiO$_2$ supports can dramatically modify the morphology, surface atomic configuration, MSI mode, and catalytic performances of Ru catalysts for CO$_2$ hydrogenation reaction, although they share the same chemical compositions. Ru NPs adhere stronger with R–TiO$_2$ than A–TiO$_2$, which disagrees with the trend that more active surface-oxygen atoms lead to stronger interfacial adhesion[28,29]. Instead, this confirms that interfacial compatibility plays critical roles in controlling the metal-support adhesion strength and MSI modes of Ru/TiO$_2$ catalysts.

For Ru/TiO$_2$ catalysts, RuO$_2$ shares the same lattice structure with R–TiO$_2$, thus, annealing RuCl$_3$–R–TiO$_2$ precursor can incorporate Ru atoms into the surface lattices of R–TiO$_2$ to form epitaxial RuO$_x$ species (Fig. 6a, b). Such interfacial RuO$_x$ species can act as anchoring layers to strongly bind Ru nanoparticles onto R–TiO$_2$ supports, which yields flat shapes with low contact angles and larger curvatures. This morphology can decrease the ratio of undercoordinated surface sites (Fig. 6a), and further modifies CO adsorptions and reaction routes. At the atomic scale, Ru$^{\delta+}$ atoms can adequately occupy Ti sites, thus bonding to R–TiO$_2$ substrate with maximized bonding strength and density. While Ru$^0$ atoms of Ru NPs can further bond to such Ru$^{\delta+}$ sites through Ru–Ru metallic bonds (Fig. 6b). This kind of binding features can minimize interfacial defects of strain, dislocation, and vacancies, thus can further highly enhance metal-support adhesion and suppress interfacial phase separation.

The lattice type of A–TiO$_2$ (I4$_1$/amd) is different from that of RuO$_2$ (P4$_2$/mnm), thus their interfacial atomic configurations do not match. Such misfit interfaces can form defects like edge dislocation and vacancy, which reduces interfacial adhesion and stability. Figure 6c–d schemes atomic interfacial contact of Ru nanoparticle on A–TiO$_2$ (101) surface. Some Ru atoms bond to surface oxygen atoms through Ru–O bonds, but their bonding lengths and strengths vary, depending on their atomic positions. Moreover, some atoms cannot effectively bond to surface-oxygen atoms due to misfit positions, and these sites form dislocations (Fig. 6d). Therefore, Ru NPs weakly adhere A–TiO$_2$ surfaces, and appear as spherical particles. This morphology yields more undercoordinated Ru sites. While the higher reducibility of surface-oxygen atoms can further drive the occurrence of SMSI effect, which further modifies CO adsorption, catalytic activity, and selectivity.

In summary, we have demonstrated that interfacial compatibility can critically control the interfacial coupling strength, surface atomic configurations, MSI modes, and catalytic performances of Ru/TiO$_2$ catalysts by varying interfacial adhesion strengths. For CO$_2$ methanation, enhanced interfacial coupling of Ru/R–TiO$_2$ can increase catalytic activity and CH$_4$ selectivity, while SMSI effect on Ru/A–TiO$_2$ can highly decrease catalytic activity and convert the product from CH$_4$ into CO. This is because Ru NPs can strongly adhere to R–TiO$_2$ supports and form flat particles with larger curvatures, in which interfacial RuO$_x$ species act as anchoring layers with R–TiO$_2$; while Ru/A–TiO$_2$ show classic SMSI effect due to lattice misfit and higher reducibility of surface oxygen atoms. Therefore, interfacial compatibility is a critical structural feature that can intrinsically modulate MSI modes and catalytic performances, which might be realized through designing interfacial atomic configurations, introducing anchoring layers, thermal annealing, and oxidation treatments. This work paves the way to improve catalytic performances through engineering interfacial compatibilities between metal NPs and supports.

## Methods

**Chemicals**. Titanium tetrachloride (TiCl$_4$, 99.5%), titanium trichloride (15.0~20% TiCl$_3$ basis in 30% HCl), TiO$_2$ (anatase, 25 nm), sodium hydroxide (NaOH, >98%), and sodium chloride (NaCl, 99.6%) were purchased from Shanghai Aladdin Biochemical Technology Co., Ltd. Ethanol (>99.7%), hydrochloric acid (HCl, 36–38%), and nitric acid (HNO$_3$, 68%) were purchased from Sinopharm Chemical Reagent Beijing Co., Ltd. Ruthenium trichloride hydrate (RuCl$_3$·xH$_2$O, 37.5–41% Ru) and titanium nitride (TiN, 99%) were purchased from Beijing Innochem Science & Technology Co., LTD.

**Syntheses of rutile-type TiO$_2$ nanorods**. R–TiO$_2$ nanorods were prepared using three methods:

(1) TiCl$_4$ (2.5 mL) was slowly added into ice water (30 mL) under vigorous stirring. After being stirred for 15 min, the solution was transferred into a 40-mL Teflon-lined stainless-steel autoclave. Then the solution was heated at 170 °C for 14 h in an oven. After cooling down to room temperature, the products were collected by centrifugation and washed 4 times with deionized water, and dried at 60 °C for 12 h[54].

(2) TiCl$_3$ (5 mL) was first mixed with 30 mL of 1.0 M NaCl solution. After being transferred into a 40 mL Teflon-lined stainless-steel autoclave, the solution was heated at 200 °C for 6 h.

(3) TiN (0.5 g) was dispersed into 30 mL of 4.0 M HNO$_3$ solution, then was transferred into a 40 mL Teflon-lined stainless-steel autoclave. The mixture was heated at 180 °C for 24 h[55].

**Syntheses of Ru/TiO$_2$ catalysts**. All Ru/TiO$_2$ catalysts were prepared following the same procedure, in which only TiO$_2$ supports were changed. (1) TiO$_2$ supports were annealed in air at 500 °C for 10 h to stabilize their surfaces. (2) Then 1.0 g of the annealed TiO$_2$ supports and a certain amount of RuCl$_3$ were mixed in 10 mL of H$_2$O under sonication for 30 min. (3) The mixtures were rapidly frozen with liquid nitrogen, and further dried in a freeze drier. (4) As-obtained powders were calcined at 400 °C for 4 h in air or directly reduced with H$_2$ to prepare Ru/TiO$_2$ catalysts.

**Characterization**. Power X-ray diffraction (XRD) data were collected on a Bruker D8 diffractometer using Cu Kα radiation (1.5418 Å), which was operated at 40 kV

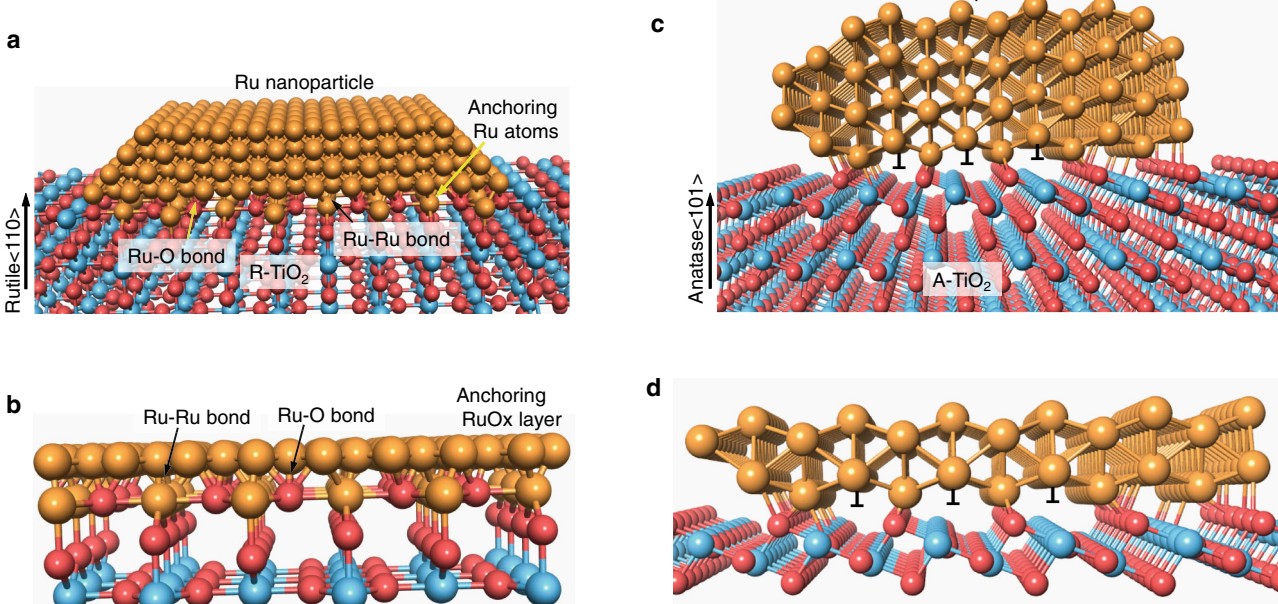

**Fig. 6 Atomic schemes showing the effects of interfacial compatibility on Ru/TiO₂ adhesion strengths. a** Scheme of epitaxial dispersion of Ru nanoparticle on rutile TiO₂(110) surface. **b** Interfacial Ru–O and Ru–Ru bonds in anchoring RuOₓ layer on R–TiO₂. **c** Scheme of Ru nanoparticle on anatase TiO₂ (101) surface. **d** Scheme of interfacial of bonds and defects of Ru with A–TiO₂.

and 40 mA with a scanning rate of 6 degree/min. High-resolution transmission electron microscopy (HRTEM), high-angle annular dark-field scanning transmission electron microscopy (HAADF-STEM), and element mapping were performed on FEI Tecnai F30 transmission electron microscope (TEM) under an acceleration voltage of 300 kV. Size distributions of Ru nanoparticles were obtained through measuring at least 100 particles. X-ray photoelectron spectroscopy (XPS) data were collected on Thermo Scientific ESCALAB 250Xi system using Al Kα line as the X-ray source. The spectra were calibrated by with C1s peak at 284.8 eV.

*H₂-TPR.* H₂ temperature-programmed reduction (H₂-TPR) and CO pulse adsorption were performed on Autochem1 II 2920 instrument. Before TPR measurement, the samples were annealed in Ar at 300 °C for 60 min, then cooled down to 50 °C. The signals were recorded online with a thermal conductivity detector (TCD), as the reactor was heated to 800 °C at a heating rate of 10 °C /min under 10% H₂–90% Ar flows.

*DRIFT.* In situ diffuse-reflectance infrared Fourier transform (DRIFT) spectra of CO adsorption were performed on Thermo Fisher Nicolet iS50 with a resolution of 4 cm⁻¹ at 25 °C. Prior to CO adsorption, the sample was treated in H₂ flow at 300 °C for 1 h and then cooled down. Prior to collecting the background spectrum, the sample was purged with Ar for 30 min. Then 5% CO/Ar flow (20 sccm) was introduced into the reactor until saturated adsorptions. DRIFT spectra were collected until no gas-phase CO could be detected with Ar purging.

*Operando FT-IR study of reaction mechanisms.* The measurements were performed on Thermo Fisher Nicolet iS50. Prior to collecting the spectra, the samples were treated in H₂ flow at 300 °C for 1 h and then purging with Ar for 1 h. The background spectrum was collected, until the reactor cooling down to 25 °C. Subsequently, the feed gas (*n*(H₂):*n*(CO₂):*n*(Ar)=15:60:25) was introduced, then the IR spectra were collected at certain temperatures after being stabilized for 30 min.

**Catalytic test of CO₂ hydrogenation.** CO₂ hydrogenation reaction was carried out in a fixed-bed reactor made of stainless steel. First, the catalysts were in situ reduced at 300 °C for 1 h with 25-sccm pure H₂ before catalytic testing. After cooling down to 100 °C, the gas was switched to the reaction gas with molar ratio of *n*(H₂):*n*(CO₂):*n*(Ar) = 60:15:25. The reaction pressure was controlled at 1 atm, and the gas hourly space velocities (GHSVs) were 12,000 mL·g⁻¹·h⁻¹. The outgassing gas compositions were detected using an online gas chromatography (GC) equipped with a TCD detector after the reactions were stabilized for 25 min at specific temperatures.

The CO₂ conversion (*X*CO₂) was calculated according to the following equation:

$$X_{CO_2} = \frac{n_{in}(CO_2) - n_{out}(CO_2)}{n_{in}(CO_2)} = 1 - \frac{A_{out}(CO_2)/A_{out}(Ar)}{A_{in}(CO_2)/A_{in}(Ar)} \quad (2)$$

where $n_{in}(CO_2)$ and $n_{out}(CO_2)$ refer to the molar number of the CO₂ before or after the reaction, respectively, the $A_{in}(CO_2)$ and $A_{in}(Ar)$ refer to the chromatographic peak areas of the CO₂ and Ar in the reaction gas, and the $A_{out}(CO_2)$ and $A_{out}(Ar)$ refer to the GC area after reactions.

The reaction rate (*v*) was calculated following:

$$v = \frac{GHSVs \times X_{CO_2} \times 15\%}{22400 \times m_{Ru}} \quad (3)$$

where GHSVs refer to the gas hourly space velocities and $m_{Ru}$ refers to the mass of the Ru. Under this condition, the products are CO and CH₄, so the selectivity of CO and CH₄ ($S_{CO}$ and $SCH_4$) meets

$$S_{CO} + S_{CH_4} = 1 \quad (4)$$

$$S_{CO} = f_{CO/CH_4} \cdot \frac{A_{CO}}{A_{CH_4}} \cdot S_{CH_4} \quad (5)$$

where the $f_{CO/CH4}$ refers to the relative correction factors of CO–CH₄ obtained by the calibrating gas; the $A_{CO}$ and $A_{CH4}$ refer to the chromatographic peak areas of CO and CH₄, respectively.

**Reporting summary.** Further information on research design is available in the Nature Research Reporting Summary linked to this article.

## Data availability

The data supporting the findings of the study are available within the paper and its Supplementary Information. Source data are provided with this paper in excel format. Source data are provided with this paper.

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

## Acknowledgements

This work was supported by the National Natural Science Foundation of China (21801012 to G.X.) and the National Key Research and Development Program of China (2018YFA0702002 to L.W.).

## Author contributions

G.X. initiated, designed, and supervised this work, and wrote the manuscript. J.Z. prepared the materials and performed catalytic tests. Z.G. carried out DRIFTS experiments. T.Z., Z.L. and W.Z. helped with preparing the materials and catalytic experiments. G.X., X.L., and L.W. supported this research. All the authors discussed the results and commented on the paper.

## Competing interests

The authors declare no competing interests.

**Additional information**

