## [Peer Review File · Nature Communications]

Title: Interfacial compatibility critically controls Ru/TiO₂ metal-support interaction modes in CO₂ hydrogenationREVIEWER COMMENTS

Reviewer #1 (Remarks to the Author):

This manuscript reports on a quite elaborated study on the effect of the structure of titania on the catalytic properties of Ru nanoparticles supported on titania. I am sorry to say that in my view this work is not meeting the requirements of significant scientific research. It is not my intention to insult the authors but there is a mayor mistake in this manuscript. The influence of the size of the Ru nanoparticles is not considered at all. The very active R-TiO₂-air-H₂ sample is very active because it has the smallest Ru particle size (See TEM data) and the primary effect is simply a higher Ru surface area per gram of catalyst (or may be more Ru-perimeter sites, if one would assume that Ru at the perimeter is the active site). Any more subtle effect caused by the interaction of metal particles with the support can only be discussed after considering this primary effect.

Other comments:

- The XPS result for R-TiO₂-air-H₂ is very likely caused by the small Ru particles as well, as small particles oxidize easier. In addition, it should be considered that the XPS experiments probes the sample after exposure to ambient.
- Activity of a catalyst is connected to the conversion level, but it is not the same thing. Activity can be calculated only based on differential experiments, or via integral description of the reactor when conversion is higher than typically 10%.
- How is ensured that the catalytic test results are not influenced by deactivation and/or mass transfer limitations?
- Selectivity can only be compared meaningful at constant conversion.
- The manuscript should also describe how the anatase samples were prepared and what the significance is of the shape of the titania crystals, including the dominant surface terminations.

Reviewer #2 (Remarks to the Author):

The manuscript deals with the investigation of the CO₂ methanation reaction over Ru/TiO₂ catalysts pretreated in H₂ or annealing in air prior to H₂ reduction. It was found that annealing RuCl₃-TiO₂ precursors in air results in a significant increase of catalytic activity for supports consisting either of anatase or rutile. Selectivity toward CH₄ is also modified by annealing RuCl₃-TiO₂ precursors in air and was found to be higher when Ru is supported on R-TiO₂ than A-TiO₂. H₂-TPR results showed that RuO_x is more stable on the surface of R-TiO₂ compared to A-TiO₂, indicating, according to the authors, a stronger interfacial coupling due to matched lattices. XPS results showed that Ru^{δ+}/Ru⁰ is higher for Ru/R-TiO₂-air-H₂ compared to Ru/A-TiO₂-air-H₂, providing evidence that the interfacial contact is higher for the former catalyst. The authors also performed FTIR experiments in order to explore the reaction pathway, which definitely requires further analysis and discussion.

Although this is an interesting study, some of the conclusions are not supported by experimental data and at many points results require more detail discussion. Therefore, in my opinion the manuscript can

be accepted for publication in the Nature Communications only after major revision, taking into account the following comments:

1. Page 14, line 252. It is written: "The result of Ru/A-TiO₂-air-H₂ is consistent with this trend, in which the sizes of Ru nanoparticles increase from 2.1 nm to 2.8 nm (Fig. S7). However, the sizes of Ru nanoparticles decrease from 2.5 nm to 1.4 nm.". The last sentence is not clear. Which is the sample, the nanoparticles of which decrease from 2.5 to 1.4 nm and under which conditions this decrease was observed?
2. In the following lines the authors suggested that the decreased ratios of Ru_{LC} sites did not result from increased particle sizes, but from the epitaxial interface contacts. Could the authors explain this more and discuss appropriately?
3. In Fig.1 it was observed that methane selectivity progressively decreases with temperature over the Ru/A-TiO₂-H₂ sample, contrary to CO₂ conversion which is progressively increases with temperature. This means that the decrease of methane selectivity should be accompanied by the appearance of another product. Otherwise CO₂ conversion should have been decreased following the same trend with methane selectivity. Did the authors detect any other product apart from CH₄? Moreover, it is not clear to me which was the reason for the observed decrease of methane selectivity.
4. In DRIFT spectra of Fig.5 it was observed that the intensity ratio of *CO/*HCO₂ decreases with increasing temperature, which according to the authors indicates that *HCO₂ can be fast reduced into *CO by H₂, but *CO accumulates on Ru surface. Based on this observation the authors concluded that the rate-determining step of CO₂ methanation reaction is the CO dissociation on Ru surface. However, in my opinion, even if CO is accumulated on Ru surface it does not necessarily mean that the rate-determining step is CO dissociation. This should be clarified in the revised manuscript.
5. In line to the above comment, if CO dissociation was the rate-determining step then bands in the $\nu(\text{CH})$ region (2800-2950 cm⁻¹) were expected to appear prior to evolution of gas phase methane. However, no such bands were detected.
6. Could the authors define the active sites for CO₂ methanation based on the results of the present study?

Reviewer #3 (Remarks to the Author):

This manuscript reports on Ru/TiO₂ catalysts used for CO₂ hydrogenation to CO and methane. The

authors find that the rutile phase of TiO₂ leads to a much more active catalyst, particularly when a calcination step is included in the catalyst synthesis. This is attributed to the more stable interface between the rutile phases of TiO₂ and RuO₂. The authors use a variety of characterization techniques to probe the interfaces; some of these are more illuminating than others. However, the XPS and CO-FTIR results are quite interesting, and overall the manuscript reports on an interesting set of findings. I recommend publication after the authors have considered the following points:

- (1) In the abstract, the word “conversion” should probably be used to replace “activity”.
- (2) Also in the abstract, it reads as if there was some change that caused an increase in activity from 31% to 89%, but it’s hard to tell what variable was changed from the context. Reading the paper, it is clear that they annealed the catalyst in air to induce the change – it is important to make this clear.
- (3) Can the authors report uncertainty values for the conversion data in Figure 1? Ideally these would be measured by determining the temperature-dependent activity for independently synthesized catalyst samples.
- (4) Regarding Figure 1(d), it’s not clear to me why there would be an enhancement effect with P25, since it is mostly anatase. Can the authors provide further explanation?
- (5) Figure 3(a) would be much more compelling if it was reported in meaningful units to determine the total amount of oxygen consumed during each reduction.
- (6) Can the authors quantify active surface area of Ru with a site-counting technique like chemisorption? This would help in understanding how much of the activity improvement after the annealing in air step on rutile is a result of the different properties of Ru active sites versus the simple abundance of them.
- (7) The authors discuss that dissociation of CO is the rate-limiting step in the reaction network. Did they carry out any experiments in which they fed CO to verify this? Do the authors have evidence that the CO₂ dissociation step (RWGS) is either kinetically irrelevant or not much affected the different types of catalysts studied?

Author Response to Reviewers' Comments

Reviewer #1:

This manuscript reports on a quite elaborated study on the effect of the structure of titania on the catalytic properties of Ru nanoparticles supported on titania. I am sorry to say that in my view this work is not meeting the requirements of significant scientific research. It is not my intention to insult the authors but there is a mayor mistake in this manuscript.

1.1 The influence of the size of the Ru nanoparticles is not considered at all. The very active Ru-TiO₂-air-H₂ sample is very active because it has the smallest Ru particle size (see TEM data) and the primary effect is simply a higher Ru surface area per gram of catalyst (or may be more Ru-perimeter sites, if one would assume that Ru at the perimeter is the active site). Any more subtle effect caused by the interaction of metal particles with the support can only be discussed after considering this primary effect.

Reply:

We sincerely thank the reviewer for this critical comment. Following this suggest we seriously consider the influence of size of Ru nanoparticles, and further characterize the catalysts with spheroidal aberration correction electron microscope to understand how Ru/TiO₂-R-air-H₂ becomes more active. The catalysts are characterized after performing CO₂ hydrogenation reactions to reflect the structures at reaction states. The results show the shapes and sizes of Ru nanoparticles are almost the same for Ru/TiO₂-R-air-H₂ and Ru/TiO₂-R-H₂. We calculate the size distributions by measuring at least 100 nanoparticles. Size distributions of Ru nanoparticles on Ru-TiO₂-air-H₂ and Ru-TiO₂-H₂ are 2.4±0.4 nm and 2.5±0.5 nm, respectively. Therefore, we may say that the enhanced catalytic performance Ru/TiO₂-R-air-H₂ does not critically result from size effects of Ru nanoparticles.

We also characterize the metal dispersion of Ru nanoparticles with CO pulse adsorption. Metal dispersions of Ru (D_{CO}) are 33.6% and 31.1% for Ru/TiO₂-R-air-H₂ and Ru/TiO₂-R-H₂, respectively, which are almost the same. The results thus indicate that size and surface area does not play critical roles in enhancing the catalytic performances of Ru nanoparticles on TiO₂-R supports. We discuss these points in the revised manuscript.

Figure 1. STEM images of Ru/TiO₂-R-air-H₂ and Ru/TiO₂-R-H₂, and their size distributions.

- 1.2 The XPS result for Ru-TiO₂-air-H₂ is very likely caused by the small Ru particles as well, as small particles oxidize easier. In addition, it should be considered that the XPS experiments probes the sample after exposure to ambient.

Reply:

We thank the reviewer for this comment. For XPS characterizations we considered the oxidation effect of Ru nanoparticles by air. After catalytic reactions, we took out the catalysts from the reaction tubes and kept them in N₂ atmosphere before XPS characterizations. As replied in comment 1.1, we verify that both the sizes and metal dispersion of Ru nanoparticles are almost the same for Ru/TiO₂-R-air-H₂ and Ru/TiO₂-R-H₂. Therefore, we insist that the different XPS signals should result from the stronger interfacial bonding of interfacial RuO_x species with rutile-TiO₂, but not from the effects of size or surface area. This effect can also be supported by the flatter shapes of Ru nanoparticles of Ru/TiO₂-R-air-H₂.

- 1.3 Activity of a catalyst is connected to the conversion level, but it is not the same thing. Activity can be calculated only based on differential experiments, or via integral description of the reactor when conversion is higher than typically 10%. How is ensured that the catalytic test results are not influenced by deactivation and/or mass transfer limitations? Selectivity can only be compared meaningful at constant conversion.

Reply:

We sincerely appreciate this helpful suggestion. In this work we devoted to demonstrating the opposite catalytic performances of Ru nanoparticles on anatase and rutile, and highlight the roles metal-support affinity in enhancing the catalytic performances of Ru nanoparticles. In our results Ru nanoparticles display opposite trends in both CO₂ conversion and CH₄ selectivity, which typically demonstrates the effects of supports on catalysis. Moreover, rutile and anatase share the same chemical composition but different lattice structures.

We are also aware that activity refers to the intrinsic catalytic capability, and can be compared at low conversions when kinetic factors like mass transfer do not play significant roles. In the catalytic tests, all the catalytic conditions were controlled the same for all samples, including the amounts of catalysts and quartz sands, flow rates, and temperature programs. We believe that kinetic factors like mass transfer limitations cannot effectively alter the conversion and selectivity for Ru nanoparticles on anatase and rutile. Moreover, the four samples also show significant differences in conversion and selectivity when CO₂ conversions are below 20%. This suggests that their different catalytic performances are not dominated by mass transfer limitations.

In the revised manuscript we have corrected the improper terms.

- 1.4 The manuscript should also describe how the anatase samples were prepared and what the significance is of the shape of the titania crystals, including the dominant surface terminations.

Reply:

We thank the reviewer for this suggestion. In the supporting information we described how the materials, including the supports and catalysts, were obtained and prepared. Rutile nanorods were prepared following a literature report. *Anatase support (25 nm) was purchased from Shanghai Aladdin Biochemical Technology Co., Ltd.* All the supports were first annealed in air at 500 °C for 10 h, thus the particles were fully ripened to avoid the effects to specific facets. Anatase nanoparticles do not display preferential facets.

Reviewer #2:

The manuscript deals with the investigation of the CO₂ methanation reaction over Ru/TiO₂ catalysts pretreated in H₂ or annealing in air prior to H₂ reduction. It was found that annealing RuCl₃-TiO₂ precursors in air results in a significant increase of catalytic activity for supports consisting either of anatase or rutile. Selectivity toward CH₄ is also modified by annealing RuCl₃-TiO₂ precursors in air and was found to be higher when Ru is supported on R-TiO₂ than A-TiO₂. H₂-TPR results showed that RuO_x is more stable on the surface of R-TiO₂ compared to A-TiO₂, indicating, according to the authors, a stronger interfacial coupling due to matched lattices. XPS results showed that Ru^{δ+}/Ru⁰ is higher for Ru/R-TiO₂-air-H₂ compared to Ru/A-TiO₂-air-H₂, providing evidence that the interfacial contact is higher for the former catalyst. The authors also performed FTIR experiments in order to explore the reaction pathway, which definitely requires further analysis and discussion. Although this is an interesting study, some of the conclusions are not supported by experimental data and at many points results require more detail discussion. Therefore, in my opinion the manuscript can be accepted for publication in the Nature Communications only after major revision, taking into account of the following comments:

Reply: We thank the reviewer for the encouraging comment and suggestion.

2.1 Page 14, line 252. It is written: "The result of Ru/A-TiO₂-air-H₂ is consistent with this trend, in which the sizes of Ru nanoparticles increase from 2.1 nm to 2.8 nm (Fig. S7). However, the sizes of Ru nanoparticles decrease from 2.5 nm to 1.4 nm.". The last sentence is not clear. Which is the sample, the nanoparticles of which decrease from 2.5 to 1.4 nm and under which conditions this decrease was observed?

Reply:

We sincerely thank the reviewer for this suggestion. In the revised manuscript we have recharacterized the catalysts with spherical aberration correction electron microscope, and calculated size distributions of Ru nanoparticles. We have rewritten this part to make the points clear to understand.

2.2 In the following lines the authors suggested that the decreased ratios of Ru_{LC} sites did not result from increased particle sizes, but from the epitaxial interface contacts. Could the authors explain this more and discuss appropriately?

Reply:

We thank the reviewer for this comment. The ratios of low-coordinated atoms usually increase as particle sizes reduce to nanometer scale owing to the limited surrounding atoms. Our results show that the projected sizes of Ru nanoparticles are almost the same for Ru/TiO₂-R-air-H₂ and Ru/TiO₂-R-H₂, but CO-DRIFTS results show that the ratio of Ru_{LC} decreases for Ru/TiO₂-R-air-H₂. This is because the shapes of Ru nanoparticles are much flatter for Ru/TiO₂-R-air-H₂ owing the stronger affinity of Ru nanoparticles with rutile supports bridged by interfacial RuO_x species. The greater curvature radius can lead to more ordered arrangement of surface atoms and increase the number of surrounding atoms. We have further explained this issue in the revised manuscript.

2.3 In Fig.1 it was observed that methane selectivity progressively decreases with temperature over the Ru/A-TiO₂-H₂ sample, contrary to CO₂ conversion which is progressively increases with temperature. This means that the decrease of methane selectivity should be accompanied by the appearance of another product. Otherwise CO₂ conversion should have been decreased following the same trend with methane selectivity. Did the authors detect any other product apart from CH₄? Moreover, it is not clear to me which was the reason for the observed decrease of methane selectivity.

Reply:

We thank the reviewer for this suggestion. We performed the catalytic CO₂ reduction reactions at normal pressures (1 atm). At this pressure the reduction products of CO₂ are usually CO and CH₄. In our experiments we just detected CO and CH₄ products. Therefore, the selectivity refers to the percentages of CH₄ and CO in the products. For Ru/A-TiO₂-H₂ sample, in addition to CH₄, CO appears as temperature increases. In the catalyst test of CO₂ hydrogenation section in Supplementary Materials, we described the calculation method of selectivity.

2.4 In DRIFT spectra of Fig.5 it was observed that the intensity ratio of *CO/*HCO₂ decreases with increasing temperature, which according to the authors indicates that *HCO₂ can be fast reduced into *CO by H₂, but *CO accumulates on Ru surface. Based on this observation the authors concluded that the rate-determining step of CO₂ methanation reaction is the CO dissociation on Ru surface. However, in my opinion, even if CO is accumulated on Ru surface it does not necessary mean that the rate-determining step is CO dissociation. This should be clarified in the revised manuscript.

Reply:

We thank the reviewer for this good comment. We have carefully reanalyzed our data and referred to similar literature reports. Many researches have figured out that *CO and formate are two possible intermediates in thermal CO₂ hydrogenation reactions. In our results, step-wise increasing reaction temperatures can lead to similar changes of *HCO₂ on the two samples (Figure 5a). This suggests that formate is not likely the intermediate, or at least not linked to the distinctly different activity on rutile. Some previous reports have also concluded that CO hydrogenation is the rate-determining step in the CO₂ hydrogenation on Ru/TiO₂ [Angew. Chem. Int. Ed. 2020, 59, 19983]. Two catalysts show different adsorption modes of ¹³CO, more obvious multicarbonyl Ru(CO)_n species at 2070 cm⁻¹ on Ru/R-TiO₂-H₂, which adsorbed on Ru_{LC} of the surfaces of Ru nanoparticles. It is inactive at low temperatures, because H₂ cannot effectively reduce it at low temperatures. We have further clarified this point in the revised manuscript.

2.5 In line to the above comment, if CO dissociation was the rate-determining step then bands in the ν(CH) region (2800-2950 cm⁻¹) were expected to be appear prior to evolution of gas phase methane. However, no such bands were detected.

Reply:

We thank the reviewer for this comment. In Figure 5 the ν(CH) region (2800-2950 cm⁻¹) was not visible because this mode is weaker than the mode at 1400-1600 cm⁻¹. By zooming in the spectra we indeed observe a mode at 2900 cm⁻¹ at 298 K as expected. This result also agrees with previous reports, thus further supports our understandings.

Figure 2. CO-DRIFT spectra at 298 K.

2.6 Could the authors define the active sites for CO₂ methanation based on the results of the present study?

Reply:

Thanks for this suggestion. In our study, the supports and Ru nanoparticles both play important roles in CO₂ methanation. Many researches have also revealed that CO₂ is first hydrogenated on the surface of the support to generate CO, and H₂ is activated on metal NPs. The *CO species diffuse onto the surfaces of metal NPs and is further hydrogenated to methane. We have discussed the point in the revised manuscript.

Reviewer #3:

This manuscript reports on Ru/TiO₂ catalysts used for CO₂ hydrogenation to CO and methane. The authors find that the rutile phase of TiO₂ leads to a much more active catalyst, particularly when a calcination step is included in the catalyst synthesis. This is attributed to the more stable interface between the rutile phases of TiO₂ and RuO₂. The authors use a variety of characterization techniques to probe the interfaces; some of these are more illuminating than others. However, the XPS and CO-FTIR results are quite interesting, and overall the manuscript reports on an interesting set of findings. I recommend publication after the authors have considered the following points:

Reply: We thank the reviewer for the encouragement.

3.1 In the abstract, the word “conversion” should probably be used to replace “activity”.

Reply:

We sincerely appreciate this suggestion. We have corrected the improper terms in the revised manuscript.

3.2 Also in the abstract, it reads as if there was some change that caused an increase in activity from 31% to 89%, but it's hard to tell what variable was changed from the context. Reading the paper, it is clear that they annealed the catalyst in air to induce the change – it is important to make this clear.

Reply:

We thank the reviewer for this suggestion. We have rewritten the sentences to make the points clearer to understand.

3.3 Can the authors report uncertainty values for the conversion data in Figure 1? Ideally these would be measured by determining the temperature-dependent activity for independently synthesized catalyst samples.

Reply:

We thank the reviewer for this suggestion. In catalysis tests we repeated the performances of all catalysts using different catalysts, and found that the performances were quite stable. The differences are less than 1% for the samples. This is because our synthetic procedures are quite easy and are the same for all catalysts. The following figure shows the temperature-dependent CO₂ conversions by independently synthesized Ru catalysts, which are almost the same as the data in Figure 1b.

Figure 3. Temperature-dependent CO₂ conversions by independently synthesized Ru catalysts.

3.4 Regarding Figure 1(d), it's not clear to me why there would be an enhancement effect with P25, since it is mostly anatase. Can the authors provide further explanation?

Reply:

We thank the reviewer for this comment. P25 is a commercial TiO₂ product composed of 80% anatase and 20% rutile. Therefore, RuO₂ can enrich on the surface of rutile phase to form epitaxial layers on P25-TiO₂, thus can further show similar enhancement effects as on pure rutile supports.

This enhancement effect had also been found in the catalytic oxidation reactions of HCl by RuO₂/P25 catalysts (see Sci. Rep., 2012, 2, 801).

3.5 Figure 3(a) would be much more compelling if it was reported in meaningful units to determine the total amount of oxygen consumed during each reduction.

Reply:

Thanks for this suggestion. TPR is a technology used to probe the reducibility of oxide species and their interfacial interaction strength with supports. The major information lies in the reduction peaks. RuCl₃ was oxidized to RuO₂ after calcination in air, thus the total amount of oxygen consumed should be equal for two samples. The ratios of different oxidation states can be compared through the areas of the peaks for each material.

3.6 Can the authors quantify active surface area of Ru with a site-counting technique like chemisorption? This would help in understanding how much of the activity improvement after the annealing in air step on rutile is a result of the different properties of Ru active sites versus the simple abundance of them.

Reply:

We thank the reviewer for this good suggestion. Following this suggestion, we perform the CO pulse adsorption to probe the active surface area of Ru nanoparticles on rutile. The results show that Ru nanoparticles display similar metal dispersions (D_{CO}), which are 33.6% and 31.1% for Ru/TiO₂-R-air-H₂ and Ru/TiO₂-R-H₂, respectively. We have also characterized the size distributions of Ru nanoparticles, which are also almost the same. These results indicate that size and surface area does not play critical roles in enhancing the catalytic performances of Ru nanoparticles on TiO₂-R supports. Therefore, we insist that the enhanced catalytic performances of Ru nanoparticles for CO₂ hydrogenation result from the altered surface configurations, which are basically modulated by the different bonding strengths of Ru nanoparticles with rutile and anatase.

sample	Ru/R-TiO ₂ -air-H ₂	Ru/R-TiO ₂ -H ₂
$D_{CO}(\%)$	33.64	31.1

3.7 The authors discuss that dissociation of CO is the rate-limiting step in the reaction network. Did they carry out any experiments in which they fed CO to verify this? Do the authors have evidence that the CO₂ dissociation step (RWGS) is either kinetically irrelevant or not much affected the different types of catalysts studied?

Reply:

We thank the reviewer for this suggestion. In FTIR of the samples, we can see that the appearance of *CO on Ru is prior to gas phase CH₄ and CO, so the reaction path follows the CO routes. Zhang et.al has also proved that *CO is the intermediate in the CO routes, and its activation on Ru surface is the rate-limiting step in the reaction network. Our result is similar to some reported results of different metals like:

(1) S. Eckle, H.-G. Anfang, R. J. Behm, J. Phys. Chem. C 2011, 115, 1361–1367;

(2) X. Wang, H. Shi, J. Szanyi, Nat. Commun. 2017, 8, 513 – 518.

(3) F. Wang, S. He, H. Chen, B. Wang, L. Zheng, M. Wei, D. G. Evans, X. Duan, J. Am. Chem. Soc. 2016, 138, 6298 – 6305.

REVIEWERS' COMMENTS

Reviewer #2 (Remarks to the Author):

I studied carefully the revised version of the manuscript with title: “Interfacial compatibility critically controls Ru/TiO₂ metal-support interaction modes in CO₂ hydrogenation”, as well as the comments of the other referee.

The authors have made sufficient corrections compared to the original draft and addressed most of my earlier concerns.

According to my opinion, the manuscript in its present form could be accepted for publication in Nature Communications.

Reviewer #3 (Remarks to the Author):

The authors have improved the paper in responding to the reviews, though in some ways their actions were fairly minimal. For example, they still do not really include an appropriate treatment of experimental uncertainty. Although their response does show the overall reproducibility of the effects they observe, they don't seem to adequately discuss or report error bars on their experiments. Per the comments of reviewer 1, the analysis of kinetics also leaves something to be desired, especially with respect to analysis of conversion versus selectivity trends. I think the results are interesting enough that I would recommend acceptance of the paper, but I do have concerns that the paper was not improved to the extent that it might have been.

Reviewer #2 (Remarks to the Author):

I studied carefully the revised version of the manuscript with title: "Interfacial compatibility critically controls Ru/TiO₂ metal-support interaction modes in CO₂ hydrogenation", as well as the comments of the other referee.

The authors have made sufficient corrections compared to the original draft and addressed most of my earlier concerns. According to my opinion, the manuscript in its present form could be accepted for publication in Nature Communications.

Reply:

Many thanks for the kind comment.

Reviewer #3 (Remarks to the Author):

The authors have improved the paper in responding to the reviews, though in some ways their actions were fairly minimal. For example, they still do not really include an appropriate treatment of experimental uncertainty. Although their response does show the overall reproducibility of the effects they observe, they don't seem to adequately discuss or report error bars on their experiments. Per the comments of reviewer 1, the analysis of kinetics also leaves something to be desired, especially with respect to analysis of conversion versus selectivity trends. I think the results are interesting enough that I would recommend acceptance of the paper, but I do have concerns that the paper was not improved to the extent that it might have been.

Reply:

Thanks a lot for the comment.